# Performance evaluation of GenXplore q4800: a novel open-platform automated nucleic acid testing system

Hsieh-Cheng Chen,[1] Jui-Chien Wu,[1] Huan-Chang Lin,[1] Yung-Tsan Chen,[1] Yu-Che Cheng,[1] Cheng-Tse Lu,[1] Keng-Tzu Fan,[1] Kai Huang,[1] Chi-Ming Wen,[1] Sung-Shih Yu,[1] Ying-Ping Lin,[1] Yen-Chin Huang,[1] Ruei-Jiun Chi[1]

**ABSTRACT** The emergence of diverse infectious diseases has led to the development of numerous molecular diagnostic kits for pathogen detection. These nucleic acid testing kits are typically categorized as manual or automated, with most automated kits requiring proprietary instruments, thereby limiting flexibility. GenXplore q4800 is a newly developed, open-platform, fully automated nucleic acid testing system designed to be compatible with a wide range of manual testing kits from different manufacturers. In this study, we evaluated the analytical and clinical performance of the GenXplore q4800 using manual molecular kits targeting sexually transmitted infections and respiratory viruses. The automated detection of inactivated pathogen controls demonstrated high repeatability, reproducibility, and sensitivity. The result of the contamination assessment was acceptable. For clinical validation, 120 urine specimens were tested for *Neisseria gonorrhoeae* and *Chlamydia trachomatis*. Results from the GenXplore q4800 were compared with those obtained using a semi-automated reference method. The GenXplore q4800 achieved 100% positive percent agreement, 100% negative percent agreement, and a Cohen's kappa coefficient of 1.0 in these two pathogens, indicating strong concordance. Additionally, feedback from 16 professional users confirmed a high level of satisfaction with the system's usability.

**IMPORTANCE** Most fully automated nucleic acid testing systems rely on proprietary reagent kits, which limit flexibility during supply shortages or emerging outbreaks. The GenXplore q4800 is a newly developed open-platform system that allows integration of third-party nucleic acid extraction and nucleic acid testing reagents, enabling laboratories to adapt quickly to diagnostic needs. The findings demonstrate that GenXplore q4800 can serve as a reliable and flexible alternative for routine molecular diagnostics, supporting more resilient laboratory workflows and expanding options beyond closed-platform systems.

**KEYWORDS** automated nucleic acid testing, sexually transmitted infections, respiratory viruses, open platform

The COVID-19 pandemic has significantly accelerated global demand for laboratory automation, highlighting its dual benefits of increasing testing throughput and reducing infection risk for laboratory personnel (1). In response, several leading manufacturers have introduced sample-to-result, or "all-in-one," automated nucleic acid amplification test (NAAT) systems, including the Abbott Alinity m system (2), Roche cobas 6800/8800 systems (3), BD MAX system (4), Cepheid GeneXpert system (5), and Hologic Panther system (6). These platforms integrate sample preparation, nucleic acid extraction, and real-time RT-PCR analysis into a single workflow. However, as closed systems, they typically require proprietary reagents and cartridges, which can limit flexibility and delay the response to emerging pathogens. By contrast, open-platform

**Peer Reviewers** Neena Abdul Abdul Hameed, Azeezia Medical College Hospital, Kollam, Kerala, India; Rohit Chawla, Maulana Azad Medical College, New Delhi, India

Address correspondence to Ruei-Jiun Chi, ray.chi@chroma.com.tw.

All authors are employees of Chroma ATE Inc., which funded this study and developed the GenXplore q4800 system. Vircell S.L. and Macherey-Nagel GmbH provided testing materials. The authors declare no other conflicts of interest.

systems offer greater adaptability by enabling laboratories to use reagents from multiple suppliers. This is especially valuable during public health emergencies when newly emerging pathogens may not yet be covered by commercial assays. In such situations, laboratories often resort to in-house protocols using conventional qPCR instruments and manually prepared reagents, particularly in resource-limited settings (7).

Automated NAAT platforms have been applied in respiratory virus detection, such as SARS-CoV-2 (4, 5), influenza A/B, and respiratory syncytial virus (RSV) (8, 9). On the other hand, they have also been extensively validated for the detection of sexually transmitted infections (STIs), including *Chlamydia trachomatis* (CT), *Neisseria gonorrhoeae* (NG), *Trichomonas vaginalis* (TV), and *Mycoplasma genitalium* (MG) (10). The automated systems demonstrated excellent sensitivity and specificity across both male and female urogenital specimens in a large multicenter study (3). These findings indicate the evolving capabilities of multiplex NAAT systems in supporting timely and comprehensive STI diagnosis. While these platforms offer rapid and reliable respiratory viruses and STI detection, they remain confined to proprietary workflows, limiting flexibility in response to changing clinical demands or supply chain disruptions. This highlights the increasing need for validated open-platform, fully automated NAAT systems that combine the advantages of reagent adaptability with automation.

Importantly, recent studies have reported non-negligible risks of false-positive results in automated NAAT workflows. Chandler et al. indicated the overall estimated false-positive rate for high-throughput NAATs was 0.04% (11). Navarathna et al. further highlighted these challenges by evaluating one of the automated systems of the SARS-CoV-2 assay and reporting an approximate false-positive rate of 3.5%, attributed not to reagent failure but to atypical amplification curves misinterpreted by the platform's automated analysis software (12). García-Salguero et al. indicated that the automated system has a notably high false-positive rate of 15.2% when compared with qRT-PCR, particularly among samples with low relative light unit values (13). Therefore, any new automated NAAT platform must demonstrate not only diagnostic accuracy and high-throughput capacity but also integrity in contamination control and interpretive reliability.

To address this need, the GenXplore q4800 was developed as an open-platform automated system that integrates liquid handling, magnetic bead-based nucleic acid extraction, auto-capping, and real-time PCR detection. The system can operate in fully automated mode or as a modular platform (e.g., extraction-only or amplification-only), offering operational flexibility for various laboratory settings. In this study, we conducted a comprehensive evaluation of the GenXplore q4800, focusing on its analytical performance—including repeatability, reproducibility, sensitivity, and cross-contamination control—as well as its clinical validation using urine specimens for CT, NG, and MG detection. Additionally, usability testing was performed with professional operators to assess workflow efficiency and user satisfaction in a practical laboratory environment.

## MATERIALS AND METHODS

### System design: GenXplore q4800 open-platform automated nucleic acid testing system

The GenXplore q4800 (Chroma ATE Inc., Taoyuan, Taiwan) is an open-platform, fully automated nucleic acid testing system that integrates the following six core modules: nucleic acid extraction, liquid handling, auto-capping, thermal cycling, optical detection, and a graphical user interface (GUI) (Fig. 1). The nucleic acid extraction module incorporates a whirl-spin bearing, heating strip, and magnetic rod to facilitate efficient specimen lysis, nucleic acid binding, bead collection, washing, vapor removal, and elution. The system is compatible with magnetic bead-based nucleic acid extraction kits, such as those from TANBead (14) and Macherey-Nagel (15), as well as manual real-time PCR kits from various manufacturers. In the subsequent analytical and clinical performance evaluation, controls and specimens will be sequentially processed through all six functional modules to complete the full system run.

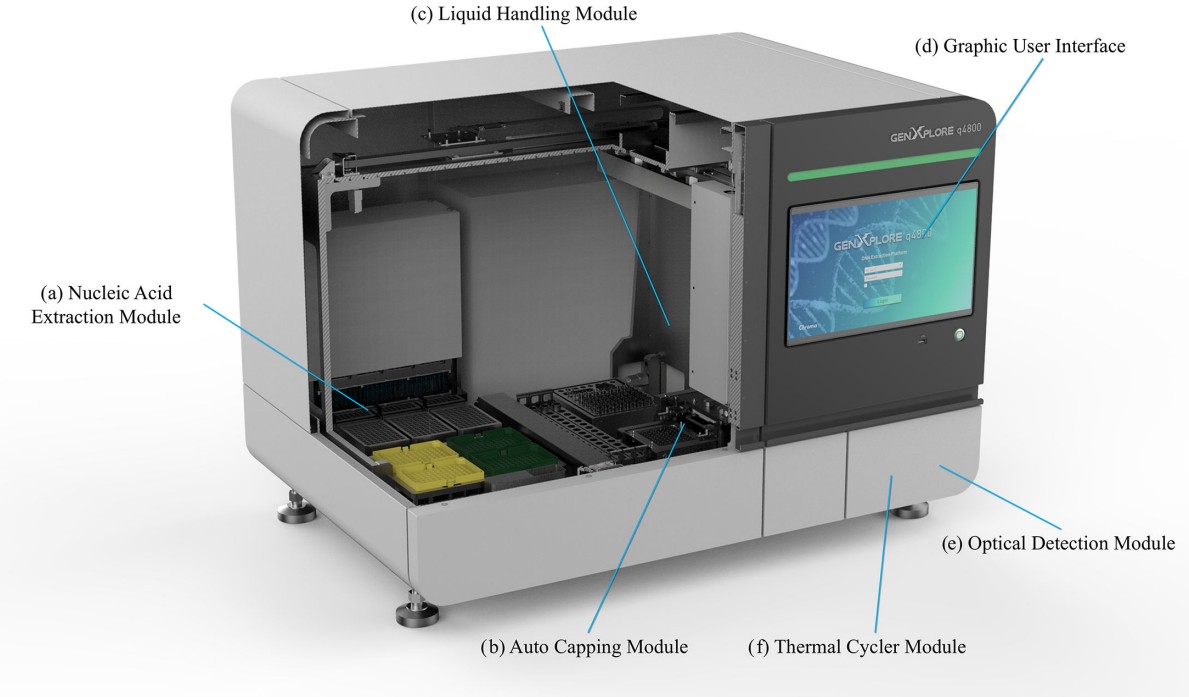

**FIG 1** Schematic design of the GenXplore q4800 automated nucleic acid testing system. (a) Nucleic acid extraction module, compatible with magnetic bead-based extraction kits. (b) Auto-capping module, compatible with validated 0.1 mL qPCR 8-strip white tubes and optical caps. (c) Liquid handling module with high dispensing accuracy (pipette tip 50 µL: coefficient of variation (CV) ≤ 10% at 5 µL; CV ≤ 2% at 50 µL and pipette tip 1,000 µL: CV ≤ 10% at 10 µL; CV ≤ 2% at 100 µL; CV ≤ 1% at 1,000 µL) and high dispensing precision (pipette tip 50 µL: CV ≤ 1.5% at 5 µL; CV ≤ 0.75% at 50 µL and pipette tip 1,000 µL: CV ≤ 3.5% at 10 µL; CV ≤ 0.75% at 100 µL; CV ≤ 0.75% at 1,000 µL). (d) GUI, designed for flexible protocol setup and intuitive operation. (e) Optical detection module supporting fluorescence detection across multiple channels: FAM, HEX/VIC/JOE, ROX/Texas Red, Cy5, and Cy5.5/Quasar 705. (f) Thermal cycler module, with temperature accuracy ≤ ±0.4℃, uniformity ≤ ±0.4℃, heating ramp rate ≥ 4℃/s, and cooling ramp rate ≥ 3℃/s. A built-in barcode scanner enables automatic sample ID recognition. The system accommodates 48 primary samples per run and 64 downstream qPCR reactions, with 16 additional wells available for controls or standards.

## Analytical performance evaluation

### Repeatability

Repeatability was assessed according to CLSI EP05-A3 (16). An assay was performed on a single instrument with 3 replicates per run, 2 runs/day, over 20 consecutive days. Key performance indicators included between-run and between-day variation across five optical channels. The OptiPure Viral Auto Plate (TANBead, Taoyuan, Taiwan) was used for nucleic acid extraction, and the SARS-CoV-2/FluA/FluB/RSV Real-Time PCR Kit (Vircell S.L., Granada, Spain) was used for amplification. An assay was conducted using 300 µL of a 10-fold diluted pooled control sample containing inactivated pathogens (Amplirun Total SARS-CoV-2/FluA/FluB/RSV Control, SWAB; Vircell S.L.).

### Reproducibility

Reproducibility was evaluated in accordance with CLSI EP05-A3 (16). Assay involved 5 replicates per run over 5 consecutive days using three separate instruments. Performance indicators included within-run, between-day, and between-instrument variation. The OptiPure Viral Auto Plate (TANBead) was used for nucleic acid extraction. Inactivated pathogen controls included Amplirun Total SARS-CoV-2/FluA/FluB/RSV Control, which is cultured from VERO E6, MDCK, and HEP-2 cell line and Amplirun Total CT/NG Control, which is formulated to simulate a human urethral exudate specimen (Vircell S.L.). Two commercial qPCR kits were evaluated: the SARS-CoV-2/FluA/FluB/RSV and CT/NG/TV/MG

Real-Time PCR kits (Vircell S.L.). Both controls included human cellular material, and the qPCR kits included detection of the human RNase P gene as an internal control.

### Analytical sensitivity

Four concentrations of pathogen controls were tested in triplicate over 3 days. The limit of detection (LoD) was defined as the lowest concentration for which all three replicates were positive. Controls used included Amplirun Total SARS-CoV-2 and Amplirun Total CT/NG (Vircell S.L.). Extraction was performed with the OptiPure Viral Auto Plate (TANBead), followed by real-time PCR with the same kits used in the reproducibility testing.

### Cross-contamination assessment

To assess the risk of cross-contamination from aerosol generation or liquid leakage, we designed a checkerboard layout using alternating high-concentration DNA and no-template controls (ddH$_2$O) across the system's full testing capacity (see Fig. 4). The high-concentration DNA sample was extracted from chicken tissue using the NucleoMag Tissue Kit (Macherey-Nagel, Düren, Germany). PCR amplification was performed using GoTaq qPCR Master Mix (Promega, WI, USA) and chicken-specific primers, consisting of 0.25 µM forward primer: 5′-TAACTTTTGTAAGCGGACACTCAT-3′ and a 0.25 µM reverse primer: 5′-GCATTACCTGCGCTGTGGC-3′.

Cross-contamination was also assessed both within and between instruments by monitoring the results of the extraction negative control (ddH$_2$O) and the qPCR negative control in each experiment. By collecting and analyzing data from these controls across repeatability, reproducibility, and analytical sensitivity assays, we were able to evaluate the potential for cross-contamination throughout the workflow.

## Clinical performance: detection of sexually transmitted pathogens

Clinical validation was conducted using 120 de-linked urine specimens obtained from a CLIA-certified laboratory (Boca Biolistics LLC, FL, USA) with IIRB (Independent Investigational Review Board, FL, USA) review. The review standard operating procedure of IIRB is consistent with the FDA guidance (17). An assay was performed at GeneRock Precision Medical Laboratory (Taoyuan, Taiwan). According to CLSI EP09-A3 (18), a minimum of 100 samples is recommended for validation. Three hundred microliters of urine specimens was tested using the GenXplore q4800 system with the OptiPure Viral Auto Plate and CT/NG/TV/MG Real-Time PCR Kit. Results were compared to a semi-automated reference method comprising the MagXtract 3200 extractor (Chroma ATE) and LightCycler 96 (Roche Diagnostics). Positive percent agreement (PPA), negative percent agreement (NPA), overall percent agreement (OPA), and Cohen's kappa coefficient were calculated from 2 × 2 contingency tables.

## Usability assessment

Based on IEC/TR 62366-2 guidelines (19), a minimum of 15 participants was used to ensure ≥91% probability of detecting at least one usability issue. Usability assessment was conducted with 16 professional users (10 laboratory technicians and 6 field application engineers) who independently operated the GenXplore q4800 and completed a standardized questionnaire. The questionnaire comprised 18 items across three categories: access control, operational friendliness, and ease of GUI setup (Table 2). Satisfaction was scored on a 100-point scale, with thresholds defined as follows: very satisfied (91–100), satisfied (81–90), average (71–80), dissatisfied (61–70), and very dissatisfied (<60).

## RESULTS

### System overview

The GenXplore q4800 integrates six functional modules: nucleic acid extraction, auto-capping, liquid handling, thermal cycling, optical detection, and a user-friendly graphical interface (Fig. 1). The system accommodates between 1 and 48 samples per run and supports overlapping runs. For example, while the first run undergoes qPCR, users can prepare samples and reagents for the next run to perform liquid handling and nucleic acid extraction concurrently. The system also supports kit and consumable traceability and is compatible with laboratory information system integration. Its open-platform design allows seamless integration of manual reagent kits into a fully automated workflow. This flexibility makes it suitable for small-sized to medium-sized laboratories seeking to consolidate various nucleic acid testing protocols. The system also supports modular use, enabling independent execution of extraction or qPCR processes.

### High repeatability across optical channels

Repeatability was assessed using inactivated controls for SARS-CoV-2, influenza A and B, and RSV, detected across five fluorescence channels over 20 consecutive days (2 runs/day). The CV for between-day and between-run analyses were as follows: FAM (SARS-CoV-2): 1.19% (day), 1.91% (run); HEX (influenza A): 1.38% (day), 2.27% (run); Texas Red (RSV): 1.04% (day), 1.94% (run); Cy5 (influenza B): 1.35% (day), 2.03% (run); Quasar 705 (internal control): 2.32% (day), 3.35% (run) (Fig. 2a). All CVs were below 5%, indicating excellent stability and repeatability of optical detection performance.

### Robust reproducibility between instruments

Reproducibility was tested across three instruments (serial numbers 69–71) over 5 days using 5 replicates per run. Two qPCR kits were evaluated with inactivated controls for respiratory viruses and STIs. The between-instrument CVs for the internal control (Quasar 705) were 1.91% and 1.25% for the two kits, respectively (Fig. 2b and c). For the pathogen targets, the between-instrument CVs were as follows: SARS-CoV-2 (FAM): 3.05%, influenza A (HEX): 2.95%, RSV (Texas Red): 3.03%, influenza B (Cy5): 4.18%, CT (FAM): 2.81%, and NG (HEX): 3.41% (Fig. S1). These results confirm high inter-instrument consistency.

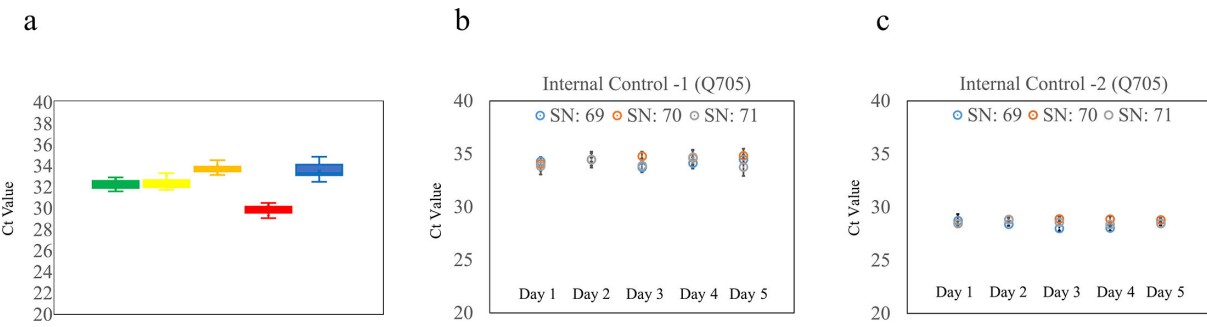

**FIG 2** Repeatability and reproducibility performance of GenXplore q4800. (a) Repeatability performance across five optical channels. The SARS-CoV-2/FluA/FluB/RSV Control (Vircell S.L.), containing four inactivated respiratory viruses and one internal control, was tested with the corresponding qPCR kit in the GenXplore q4800 over 20 days (40 runs, 2 runs/day). The fluorescence signals were detected across five channels: FAM (SARS-CoV-2, green box), HEX (influenza A, yellow box), Texas Red (RSV, orange box), Cy5 (influenza B, red box), and Quasar 705 (internal control, blue box). Box plots show Ct value distributions for each optical channel, demonstrating low variability across 20 days. (b and c) Inter-instrument reproducibility of internal control Ct values. Reproducibility was assessed using three instruments (serial numbers 69–71) over 5 days, with 5 replicates per day. Internal control Ct values from two qPCR kits were plotted for each instrument: serial numbers 69 (blue), 70 (orange), and 71 (gray). Error bars represent ±1 SD. The data confirm excellent consistency among instruments.

## High analytical sensitivity for pathogen detection

Analytical sensitivity was determined using serial dilutions of pathogen controls over 3 days, with 3 replicates per concentration. The GenXplore q4800 achieved detection thresholds lower than those claimed in the respective kit instructions for use (IFU): SARS-CoV-2: 2 copies/reaction (vs. IFU: 4), CT: 2.2 copies/reaction (vs. IFU: 13), and NG: 0.9 copies/reaction (vs. IFU: 1.8) (Fig. 3). These results demonstrate high analytical sensitivity when paired with compatible manual qPCR kits.

## No evidence of cross-contamination

To evaluate system integrity and contamination control, a checkerboard layout of high-concentration DNA (1 and 10 ng/µL) and no-template controls was used across 64 qPCR vials. The full workflow—including primary sample loading, nucleic acid extraction, eluate transfer, and PCR setup—was executed automatically. All no-template vials tested negative (0/32), while all DNA-positive vials yielded consistent amplification (32/32, average Ct: 19.7, CV: 1.27%) (Fig. 4). This confirms that the system is free from detectable cross-contamination under high-throughput operating conditions.

In the repeatability assay, all 40 extraction negative controls and 40 qPCR negative controls tested negative across five optical channels. Similarly, all 15 extraction and 15 qPCR negative controls in the reproducibility assay, and all 3 extraction and 3 qPCR negative controls in the analytical sensitivity assay, also tested negative. These results confirm the absence of detectable cross-contamination within and between instruments.

## Excellent agreement with reference method in clinical testing

Clinical performance was evaluated using 120 urine specimens tested for CT, NG, and MG. Results from the GenXplore q4800 were compared with those obtained from a semi-automated reference method (MagXtract 3200 combined with the LightCycler 96). For CT, the agreement indicators were as follows: PPA 100% (95% CI: 95.86%–100%), NPA 100% (95% CI: 88.75%–100%), OPA 100% (95% CI: 96.89%–100%), and Cohen's kappa 1.0. For NG: PPA 100% (95% CI: 84.50%–100%), NPA 100% (95% CI: 96.26%–100%), OPA 100% (95% CI: 96.89%–100%), and Cohen's kappa 1.0. For MG: PPA 100% (95% CI: 60.97%–100%), NPA 100% (95% CI: 96.74%–100%), OPA 100% (95% CI: 96.89%–100%), and Cohen's kappa 1.0 (Table 1). Although MG results also showed complete agreement, the number of MG-positive specimens ($n = 6$) was limited, which reduces the statistical robustness for this pathogen. Nevertheless, these findings demonstrate strong concordance between the GenXplore q4800 and the reference method, particularly for CT and NG.

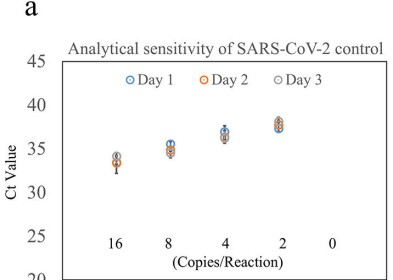 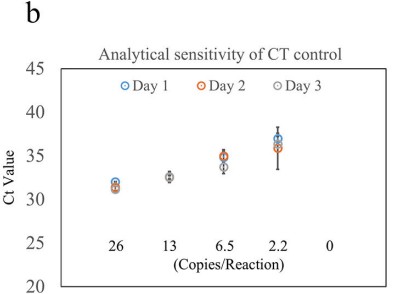 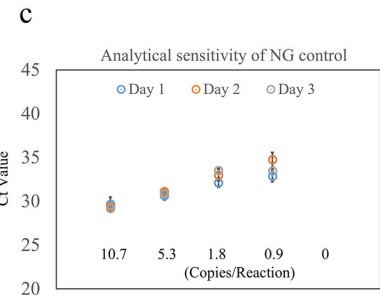

**FIG 3** Analytical sensitivity of GenXplore q4800 for detecting SARS-CoV-2, CT, and NG. Three inactivated pathogen controls (SARS-CoV-2, CT, and NG) were tested across four concentrations in triplicate over 3 days. (a) LoD of SARS-CoV-2 met IFU claims (4 copies/reaction). (b) LoD of CT met IFU claims (13 copies/reaction). (c) LoD of NG met IFU claims (1.8 copies/reaction). The data demonstrate high sensitivity for low-copy pathogen detection.

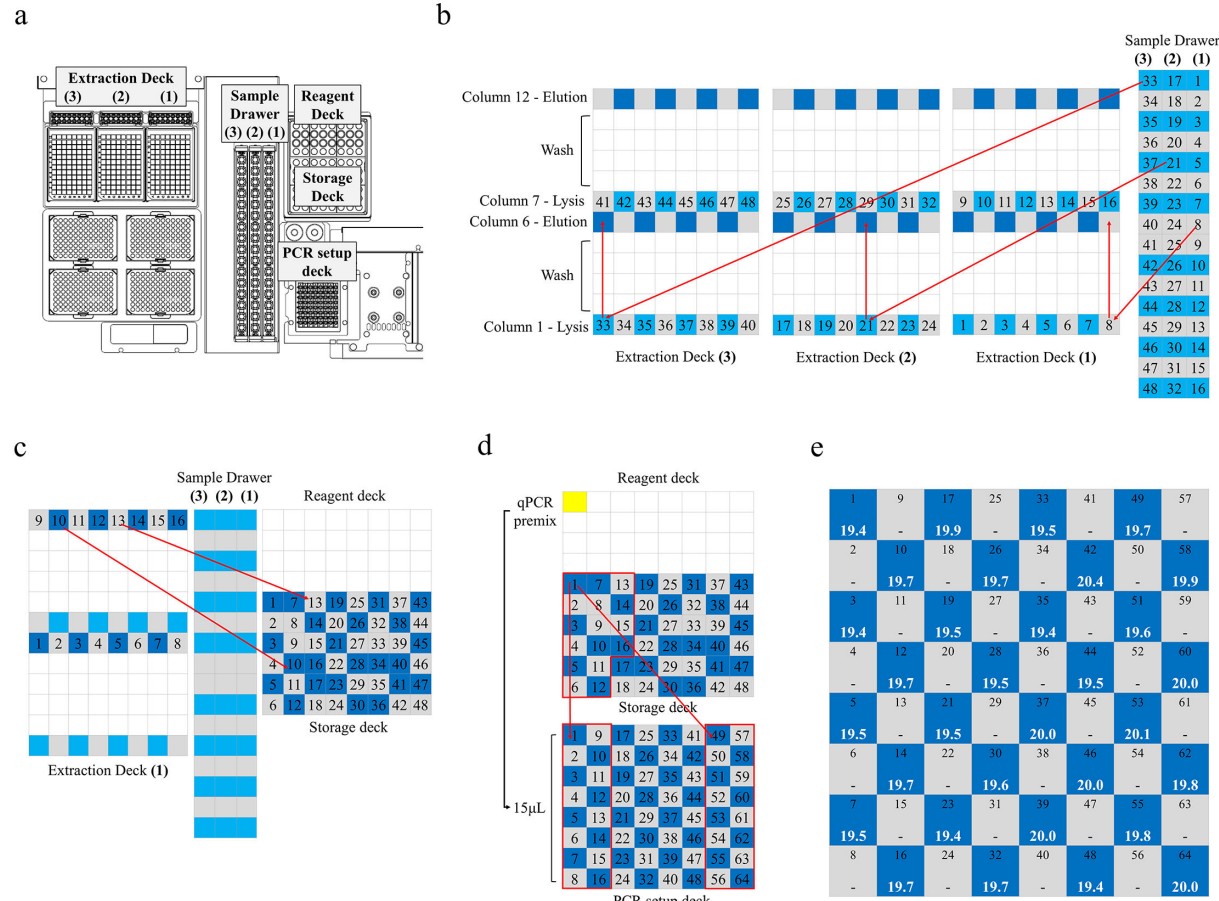

**FIG 4** Workflow layout in GenXplore q4800 for contamination assessment and final qPCR results. (a) Layout of the system's workstation. The possible area causing contamination consists of extraction deck, sample drawer, reagent deck, storage deck, and PCR setup deck. (b) Movement path of the automated pipettor from the sample drawer to the extraction deck (column 1) and the movement path of the nucleic acid extraction module, including lysis, wash, and elution steps. Color grid: light blue = 1 ng/μL DNA, dark blue = 10 ng/μL DNA, and gray = ddH$_2$O. (c) Movement path of the automated pipettor from elution wells to storage tubes to transfer eluate. (d) Movement path of the automated pipettor from storage tubes to PCR setup deck. Fifteen microliters of qPCR premix and DNA template will be dispensed into qPCR vials. Storage samples 1–16 will be dispensed into qPCR vials 1–16 and qPCR vials 49–64. The red frame indicates mapping between storage tubes and qPCR vials. (e) Dark blue grids indicate qPCR vials containing high-concentration (50 ng/vial) DNA, while the white numbers represent Ct values. Gray grids indicate no-template control qPCR vials. All 32 positive vials showed successful amplification (mean Ct: 19.7 and CV: 1.27%), while all 32 no-template vials were negative, confirming zero cross-contamination.

## Positive user experience and system usability

Sixteen professional users who participated in a usability survey were independent and affiliated with either healthcare institutions or potential distribution partners. Most had prior experience operating similar automated instruments. These participants completed the survey after hands-on experience with the GenXplore q4800. The questionnaire covered access control, operational workflow, and GUI design (Table 2). The overall average satisfaction score was 87.6, and the average score ranges for laboratory technicians and field application engineers were 80.2–95.2 and 82.1–88.9, respectively. All categories averagely scored above 80, indicating high levels of satisfaction across the board.

## DISCUSSION

The closed, all-in-one, nucleic acid testing systems have been developed and commercialized by major diagnostic companies in recent years (20). While these systems offer convenience, high throughput, and integrated automation, their reliance on proprietary

**TABLE 1** Clinical performance comparison with reference method[a]

| Detection | GenXplore q4800 | MagXtract 3200 with LightCycler 96 | | Total |
| --- | --- | --- | --- | --- |
| | | **Positive** | **Negative** | |
| CT (FAM) | Positive | 89 | 0 | 89 |
| | Negative | 0 | 31 | 31 |
| | Total | 89 | 31 | 120 |
| NG (HEX) | Positive | 21 | 0 | 21 |
| | Negative | 0 | 99 | 99 |
| | Total | 21 | 99 | 120 |
| MG (ROX) | Positive | 6 | 0 | 6 |
| | Negative | 0 | 114 | 114 |
| | Total | 6 | 114 | 120 |

[a]Contingency tables for 120 urine specimens comparing GenXplore q4800 results with a semi-automated reference system (MagXtract 3200 combining with LightCycler 96) for detecting CT (FAM), NG (HEX), and *Mycoplasma genitalium* (MG; ROX). In all three cases, GenXplore q4800 achieved 100% agreement with the reference method and a Cohen's kappa coefficient of 1.0.

reagents significantly limits flexibility (2–6). During outbreaks of emerging pathogens, laboratories are often forced to return to in-house assays or manual qPCR protocols, as new commercial kits may not be immediately available.

The GenXplore q4800 was designed to overcome existing limitations by providing a fully automated, open-platform solution capable of integrating manual nucleic acid extraction and amplification kits from multiple manufacturers. Because of its open-platform design, the total turnaround time largely depends on the specific reagent kits used. In this study, processing a full 48-sample single run required 3.0 to 4.5 hours. While many closed systems are optimized for high-throughput processing, the GenXplore q4800 is not intended to outperform them in terms of volume. For comparison, the Alinity m system can process up to 300 samples in 8 hours, the cobas 6800 system handles 96 tests in approximately 3.5 hours, and the Panther system processes 300 samples in 8 hours. Although the BD MAX system has a throughput comparable to the GenXplore q4800, it processes only 24 results in approximately 3 hours.

Our findings demonstrate that the GenXplore q4800 provides high precision and stability across all optical detection channels, with CV for both repeatability and reproducibility consistently below 5%. The system also showed excellent analytical sensitivity, achieving a lower LoD for SARS-CoV-2, CT, and NG compared to those reported in the manufacturers' IFU. Although the analytical sensitivity evaluation was based on three replicates—fewer than the 20 replicates recommended by CLSI EP17-A2 (21)—the qPCR kits used were CE-*in vitro* diagnostic (CE-IVD) certified, indicating that their LoD had been previously validated during reagent product registration. According to the IFU of qPCR kits, the LoD is determined using 20 replicates per dilution, with ≥95% positivity at the lowest detectable concentration. While the limited number of replicates in our GenXplore q4800 evaluation may reduce statistical robustness, the system demonstrated high reproducibility (Fig. 2), and the three replicates of LoD results by the GenXplore q4800 outperformed the IFU claimed. These findings support the reliability of the GenXplore q4800 when used with certified reagents.

No cross-contamination was observed based on the results of extraction and qPCR negative controls in both within-run and between-run experiments, despite each run including fewer than 20 tests. Even under high-throughput, full-capacity conditions, the system demonstrated no evidence of cross-contamination, although the number of replicates was limited. In the inter-instrument reproducibility study, three pathogen controls tested on instrument serial number 69 showed slightly lower Ct values (Fig. S1d through f). However, the internal control results exhibited CV below 2%, indicating high system stability. The slight Ct shifts are therefore likely attributable to control-specific batch effects rather than instrument-related variability. Importantly, the observed differences—within an inter-instrument CV of <5%—are not expected to impact diagnostic interpretation.

TABLE 2  Usability evaluation based on professional user feedback[a]

| Group | Number | Items | Average satisfied scores |
|---|---|---|---|
| Access control | 1 | The access interface only allows authorized personnel to login | 93.5 |
| Operational friendliness | 2 | Sample drawer can firmly place 48 specimens, and the drawer can stand steadily on the lab work bench | 89.4 |
| | 3 | Sample drawer can be easily placed into the instrument | 88.2 |
| | 4 | All barcodes on the primary sample tube can be scanned successfully | 82.4 |
| | 5 | The extraction kits can be correctly placed into three decks in the instrument | 82.9 |
| | 6 | The PCR reagent (premix) can be correctly placed into the instrument | 90.3 |
| | 7 | The consumable (spin tip) can be correctly placed into the instrument | 88.9 |
| | 8 | The consumable (pipette tip) can be correctly placed into the instrument | 91.3 |
| | 9 | The consumable (storage tube) can be correctly placed into the instrument | 87.8 |
| | 10 | The consumable (qPCR cap) can be correctly placed into the instrument | 81.9 |
| | 11 | The consumable (qPCR vial) can be correctly placed into the instrument | 91.1 |
| | 12 | The alarm for work completion can be obviously noticed | 88.8 |
| | 13 | Waste channel and trash bag can be easily installed to instrument | 89.3 |
| Ease of GUI setup | 14 | The independent function can be clearly chosen and set up in the GUI | 87.4 |
| | 15 | The decontamination (UV light/HEPA) function can be clearly chosen and turned on in the GUI | 87.3 |
| | 16 | The cooling function can be clearly chosen and set up in the GUI | 82.8 |
| | 17 | The experimental result can be output to user's laptop | 86.5 |
| | 18 | The new protocol can be imported to the instrument | 86.5 |
| | | Overall AVG | 87.6 |

[a]Eighteen items covering access control, operational friendliness, and ease of GUI setup were scored by 16 users. Average satisfaction scores exceeded 80 across all categories, with an overall average of 87.6, indicating high system usability.

Clinical validation further demonstrated the reliability of the GenXplore q4800. Among 120 urine specimens tested for CT, NG, and MG, results showed high concordance with those obtained using a semi-automated reference method. Although the number of MG-positive specimens was limited, the strong agreement observed for CT and NG provides compelling evidence of the system's robustness in clinical diagnostic applications.

Beyond performance, usability is a critical factor for laboratory adoption. Feedback from 16 professional users—including both laboratory technicians and field application engineers—highlighted strong satisfaction with the system's access control, operational workflow, and graphical interface. The overall usability score averaged 87.6 out of 100, confirming that the system meets practical user needs in addition to technical performance requirements.

The limitations of this study include the limited number of replicates used for the LoD determination and cross-contamination assessment under high-throughput conditions. In addition, the small number of MG-positive cases may have reduced statistical power, and clinical validation was performed using only urine specimens. Future studies are planned to expand the applications of the GenXplore q4800 to include additional specimen types and a broader range of pathogen targets.

The deployment of this novel open-platform system can follow different pathways depending on the end-user type. For *in vitro* diagnostic end users, validated system solutions may be delivered through local distributors holding medical device licenses—many of whom already supply various brands of manual reagent kits. These distributors can offer pre-validated workflows tailored to the GenXplore q4800 platform.

For laboratory-developed test users, who typically operate without distributor support, implementation would require close collaboration. In these cases, the system must be validated in combination with the laboratory's in-house reagents, with

optimization of automation parameters performed jointly to ensure reliable performance.

## Conclusion

The GenXplore q4800, a fully automated open-platform nucleic acid testing system, demonstrated robust analytical and clinical performance while maintaining operational flexibility. The system provided high precision, repeatability, and reproducibility across optical detection channels and achieved high sensitivity in the detection of inactivated controls. No evidence of cross-contamination was observed, even under high-throughput stress conditions. Clinical testing of 120 urine specimens showed strong agreement with a semi-automated reference method for CT and NG. Furthermore, usability testing with 16 professional users demonstrated high levels of satisfaction, confirming the system's practical suitability for routine diagnostic laboratories.

## Future perspectives

The ability of the GenXplore q4800 to integrate manual reagent kits from multiple manufacturers provides laboratories with a flexible and scalable alternative to closed systems. Broader clinical validation and expansion to syndromic and high-throughput testing applications could further enhance its clinical utility and impact in modern molecular diagnostic workflows.

## AUTHOR AFFILIATION

[1]Life Science Instrument Development Center, Chroma ATE Inc., Taoyuan, Taiwan

## AUTHOR ORCIDs

Hsieh-Cheng Chen  http://orcid.org/0000-0001-6485-3270
Ruei-Jiun Chi  http://orcid.org/0009-0004-8092-5202

## AUTHOR CONTRIBUTIONS

Hsieh-Cheng Chen, Conceptualization, Formal analysis, Investigation, Methodology, Validation | Jui-Chien Wu, Conceptualization, Data curation, Formal analysis, Investigation, Project administration, Software, Validation | Huan-Chang Lin, Conceptualization, Formal analysis, Investigation, Methodology, Validation | Yung-Tsan Chen, Formal analysis, Investigation, Methodology, Software, Validation | Yu-Che Cheng, Conceptualization, Formal analysis, Investigation, Software, Validation | Cheng-Tse Lu, Formal analysis, Investigation, Validation | Keng-Tzu Fan, Investigation, Project administration, Visualization | Kai Huang, Conceptualization, Investigation, Validation | Chi-Ming Wen, Conceptualization, Investigation, Validation | Sung-Shih Yu, Investigation, Validation | Ying-Ping Lin, Data curation, Investigation, Software | Yen-Chin Huang, Investigation, Project administration | Ruei-Jiun Chi, Conceptualization, Investigation, Project administration, Resources, Software, Supervision

## ADDITIONAL FILES

The following material is available online.

### Supplemental Material

**Figure S1 (Spectrum03092-25-S0001.tif).** Inter-instrument reproducibility of Ct values for pathogen targets.

Open Peer Review

**PEER REVIEW HISTORY (review-history.pdf).** An accounting of the reviewer comments and feedback.

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
