## [Reviewer comments · Microbiology Spectrum]

Microbiology Spectrum

Performance Evaluation of GenXplore® q4800: A Novel Open-Platform Automated Nucleic Acid Testing System

Hsieh-Cheng Chen, Jui-Chien Wu, Huan-Chang Lin, Yung-Tsan Chen, Yu-Che Cheng, Cheng-Tse Lu, Keng-Tzu Fan, Kai Huang, Chi-Ming Wen, Sung-Shih Yu, Ping-Ying Lin, Yen-Chin Huang, and Ruei-Jiun Chi

Corresponding Author(s): Ruei-Jiun Chi, Chroma ATE Inc.

Review Timeline:

Submission Date:	September 25, 2025
Editorial Decision:	October 29, 2025
Revision Received:	November 17, 2025
Accepted:	December 7, 2025

Editor: Benjamin Liu

Reviewer(s): Disclosure of reviewer identity is with reference to reviewer comments included in decision letter(s). The following individuals involved in review of your submission have agreed to reveal their identity: Neena Abdul Abdul Hameed (Reviewer #1); Rohit Chawla (Reviewer #4)

Transaction Report:

DOI: <https://doi.org/10.1128/spectrum.03092-25>

Re: Spectrum03092-25 (Performance Evaluation of GenXplore® q4800: A Novel Open-Platform Automated Nucleic Acid Testing System)

Dear Dr. Ruei-Jiun Chi:

Thank you for the privilege of reviewing your work. Below you will find my comments, instructions from the Spectrum editorial office, and the reviewer comments.

Editor's comments:

Discussion: Lines 225-226: "The closed, all-in-one, nucleic acid testing systems have been developed and commercialized by major diagnostic companies in recent years. ": There are no references to support this statement. More references should be cited, with this one (Liu, B.M. Isothermal nucleic acid amplification technologies and CRISPR-Cas based nucleic acid detection strategies for infectious disease diagnostics. p 30-47. In Manual of Molecular Microbiology: Fundamentals and Applications; ASM Press: Washington, DC, USA, 2025.doi:10.1128/9781683674597.ch03.) as an example (citing is optional).

Please return the manuscript within 30 days; if you cannot complete the modification within this time period, please contact me. If you do not wish to modify the manuscript and prefer to submit it to another journal, notify me immediately so that the manuscript may be formally withdrawn from consideration by Spectrum.

Revision Guidelines

Sincerely,
Benjamin Liu
Editor
Microbiology Spectrum

Reviewer #1 (Comments for the Author):

Although this is a novel study, it requires more data regarding the other extraction methods
References are missing in the discussion part. Kindly elaborate on it

Reviewer #3 (Comments for the Author):

The authors evaluated, a newly developed open-platform, fully automated nucleic acid testing system, for analytical and clinical performance using third-party molecular diagnostic kits targeting sexually transmitted infections and respiratory viruses. The study is well structured, with methodology aligned to CLSI standards and appropriate statistical analysis, including PPA, NPA, and Cohen's kappa. The work demonstrates strong analytical reliability with excellent repeatability, reproducibility, sensitivity, and absence of cross-contamination. Clinical validation showed perfect agreement with a semi-automated reference method, confirming diagnostic accuracy. The inclusion of usability testing adds practical relevance, and overall presentation is clear and systematic.

However, the evaluation could have been strengthened with broader validation. The pathogen panel (CT, NG, MG) and specimen type (urine) are somewhat limited, and the small number of Mycoplasma genitalium-positive samples reduces statistical power. The contamination control assessment, although showing no observed contamination in a single checkerboard setup, could have been more robust with additional runs or inter-instrument studies. Also, analytical sensitivity (LoD) testing involved fewer replicates than recommended by CLSI EP17-A2, and a higher replicate number per dilution would have provided stronger performance validation. Inclusion of additional pathogens, specimen types, and standardized reagent timing would have further enhanced the generalizability and comparative value.

The following aspects should be addressed to strengthen the manuscript:

1. A clear conflict of interest and funding statement should be included, given all authors are affiliated with the instrument development center.
2. The usability assessment involved 16 professional users, but the manuscript does not specify whether these participants were independent, had prior experience with the system, or were affiliated with the manufacturer. Clarifying these aspects would strengthen the credibility of the user satisfaction findings.
3. The Discussion could be strengthened by briefly comparing the reported run time (3-4.5 hours) and overall cost considerations with established closed systems, to better contextualize the platform's practical advantages and limitations.
4. The manuscript specifies that the system accommodates 48 samples per run. However, it does not indicate whether overlapping runs, consumable tracking, or random-access loading are supported. A brief clarification of workflow flexibility would help better understand throughput capacity in practical settings

Reviewer #4 (Comments for the Author):

The authors have presented data on performance characteristics' of GenXplore® q4800, a newly developed, open-platform, fully automated nucleic acid testing system designed to be compatible with a wide range of manual testing kits from different manufacturers.

The authors have mentioned that the repeatability studies were carried out by the use of inactivated controls for various pathogens. The authors are requested to clarify whether these controls were sequentially processed through all the six functional modules of the system.

The authors have mentioned that one of the criteria to analyze the reproducibility was to calculate the percentage CV of the internal control between instruments, however, the details of the internal control has not been provided.

GenXplore® q4800 has been pitched as an open-platform which can be used with various molecular diagnostic kits for pathogen detection. However, the laboratory would need to perform validation studies for each molecular diagnostic kit, to establish its performance characteristics, prior to use on patient samples, which may be beyond the capacity of many laboratories. The authors are requested to comment on how this issue can be addressed.

[revised manuscript text omitted]

comprehensively evaluated the system's analytical and clinical performance, as well as
its usability in real-world laboratory settings.

Our findings demonstrate that the GenXplore[®] q4800 provides high precision and
stability across all optical detection channels, with coefficients of variation (CVs) for
both repeatability and reproducibility consistently below 5%. The system also showed
excellent analytical sensitivity, achieving lower limits of detection (LoDs) for SARS-
CoV-2, Chlamydia trachomatis (CT), and Neisseria gonorrhoeae (NG) compared to those
reported in the manufacturers' instructions for use (IFUs). Although the analytical
sensitivity evaluation was based on three replicates—fewer than the 20 replicates
recommended by CLSI EP17-A2—the qPCR kits used were CE-IVD certified, indicating
that their LoDs had been previously validated while reagent product registration. Given
the high reproducibility demonstrated by the GenXplore[®] q4800, we believe the observed
sensitivity reflects reliable system performance when integrated with certified reagents.

No cross-contamination was observed, even under high-throughput conditions
specifically designed to challenge the system's workflow integrity. In the inter-
instrument reproducibility study, three pathogen controls tested on instrument serial no.
69 showed slightly lower Ct values (Figure S1d–S1f). However, the internal control
results exhibited coefficients of variation (CVs) below 2%, indicating high system
stability. The slight Ct shifts are therefore likely attributable to control-specific batch
effects rather than instrument-related variability. Importantly, the observed differences—
within an inter-instrument CV of less than 5%—are not expected to impact diagnostic
interpretation.

[revised manuscript text omitted]

1c. Liquid handling module with high dispensing accuracy (Pipette tip 50 μ L: CV \leq 10%
at 5 μ L; CV \leq 2% at 50 μ L. Pipette tip 1000 μ L: CV \leq 10% at 10 μ L; CV \leq 2% at 100 μ L;
CV \leq 1% at 1000 μ L) and high dispensing precision (Pipette tip 50 μ L: CV \leq 1.5% at 5
μ L; CV \leq 0.75% at 50 μ L. Pipette tip 1000 μ L: CV \leq 3.5% at 10 μ L; CV \leq 0.75% at 100
μ L; CV \leq 0.75% at 1000 μ L)

1d. Graphical user interface (GUI), designed for flexible protocol setup and intuitive
operation.

1e. Optical detection module supporting fluorescence detection across multiple channels:
FAM, HEX/VIC/JOE, ROX/Texas Red, Cy5 and Cy5.5/Quasar 705.

1f. Thermal cycler module, with temperature accuracy $\leq \pm 0.4$ $^{\circ}$ C, uniformity $\leq \pm 0.4$ $^{\circ}$ C,
heating ramp rate ≥ 4 $^{\circ}$ C/s, and cooling ramp rate ≥ 3 $^{\circ}$ C/s.

A built-in barcode scanner enables automatic sample ID recognition. The system
accommodates 48 primary samples per run and 64 downstream qPCR reactions, with 16
additional wells available for controls or standards.

**Figure 2. Repeatability and reproducibility performance of GenXplore® q4800.**

2a. Repeatability performance across five optical channels. The SARS-CoV-
2/FluA/FluB/RSV Control (Vircell S.L.), containing four inactivated respiratory viruses
and one internal control, was tested with the corresponding qPCR kit in the GenXplore®
q4800 over 20 days (40 runs, 2 runs/day). The fluorescence signals were detected across
five channels: FAM (SARS-CoV-2, green box), HEX (Influenza A, yellow box), Texas
Red (RSV, orange box), Cy5 (Influenza B, red box), and Quasar 705 (internal control,
blue box). Box plots show Ct value distributions for each optical channel, demonstrating
low variability across 20 days.

2b & 2c. Inter-instrument reproducibility of internal control Ct values. Reproducibility
was assessed using three instruments (serial numbers 69–71) over five days, with five
replicates per day. Internal control Ct values from two qPCR kits were plotted for each
instrument: serial no. 69 (blue), 70 (orange), and 71 (gray). Error bars represent ± 1 SD.
The data confirm excellent consistency among instruments.

(2a)

388

(2b)

(2c)

**Figure 3. Analytical sensitivity of GenXplore® q4800 for detecting SARS-CoV-2, CT,**
 **and NG.**

Three inactivated pathogen controls (SARS-CoV-2, *Chlamydia trachomatis*, and
 *Neisseria gonorrhoeae*) were tested across four concentrations in triplicate over three
 399 days. 3a. LoD of SARS-CoV-2 met IFU claims (4 copies/reaction). 3b. LoD of CT met
 IFU claims (13 copies/reaction). 3c. LoD of NG met IFU claims (1.8 copies/reaction).
 The data demonstrate high sensitivity for low-copy pathogen detection.

(3a)

(3b)

(3c)

**Figure 4. Workflow layout in GenXplore[®] q4800 for contamination assessment and**
**final qPCR results.**

4a. Layout of the system's workstation. The possible area causing contamination consists
of extraction deck, sample drawer, reagent deck, storage deck and PCR setup deck.

4b. Movement path of the automated pipettor from the sample drawer to the extraction
deck (column 1) and the movement path of the nucleic acid extraction module, including
lysis, wash, and elution steps. Color grid: Light blue = 1 ng/ μ L DNA; Dark blue = 10
427 ng/ μ L DNA; Gray = ddH₂O.

4c. Movement path of the automated pipettor from elution wells to storage tubes to
transfer eluate.

4d. Movement path of the automated pipettor from storage tubes to PCR setup deck. 15
μ L of premix and DNA template will be dispensed into qPCR vials. Storage sample 1 to
16 will be dispensed into qPCR vial 1 to 16 and qPCR vial 49 to 64. The red frame
indicates mapping between storage tubes and qPCR vials.

4e. Dark blue grids indicate qPCR vials containing high-concentration (50ng/vial) DNA;
white numbers represent Ct values. Gray grids indicate no-template control qPCR vials.
All 32 positive vials showed successful amplification (mean Ct: 19.7, CV: 1.27%), while
all 32 no-template vials were negative, confirming zero cross-contamination.

(4a)

(4b)

(4c)

(4d)

(4e)

1	9	17	25	33	41	49	57
19.4	-	19.9	-	19.5	-	19.7	-

2	10	18	26	34	42	50	58
-	19.7	-	19.7	-	20.4	-	19.9

3	11	19	27	35	43	51	59
19.4	-	19.5	-	19.4	-	19.6	-

4	12	20	28	36	44	52	60
-	19.7	-	19.5	-	19.5	-	20.0

5	13	21	29	37	45	53	61
19.5	-	19.5	-	20.0	-	20.1	-

6	14	22	30	38	46	54	62
-	19.7	-	19.6	-	20.0	-	19.8

7	15	23	31	39	47	55	63
19.5	-	19.4	-	20.0	-	19.8	-

8	16	24	32	40	48	56	64
-	19.7	-	19.7	-	19.4	-	20.0

**Table 1. Clinical performance comparison with reference method.**

Contingency tables for 120 urine specimens comparing GenXplore[®] q4800 results with a
 semi-automated reference system (MagXtract[®] 3200 combining with LightCycler[®] 96)
 for detecting: 1a. *Chlamydia trachomatis* (CT, FAM); 1b. *Neisseria gonorrhoeae* (NG,
 HEX); 1c. *Mycoplasma genitalium* (MG, ROX). In all three cases, GenXplore[®] q4800
 achieved 100% agreement with the reference method and a Cohen's kappa coefficient of
 1.0.

(1a)

CT (FAM)		MagXtract [®] 3200 with LightCycler [®] 96		
		Positive	Negative	Total
GenXplore [®] q4800	Positive	89	0	89
	Negative	0	31	31
	Total	89	31	120

(1b)

NG (HEX)		MagXtract [®] 3200 with LightCycler [®] 96		
		Positive	Negative	Total
GenXplore [®] q4800	Positive	21	0	21
	Negative	0	99	99
	Total	21	99	120

(1c)

MG (ROX)		MagXtract [®] 3200 with LightCycler [®] 96		
		Positive	Negative	Total
GenXplore [®] q4800	Positive	6	0	6
	Negative	0	114	114
	Total	6	114	120

**Table 2. Usability evaluation based on professional user feedback.**

Eighteen items covering access control, operational friendliness, and ease of GUI setup
 were scored by 16 users. Average satisfaction scores exceeded 80 across all categories,
 with an overall average of 87.6, indicating high system usability.

Group	Number	Items	Average Satisfied Scores	
Access control	1	The access interface only allows authorized personnel to login	93.5	
	2	Sample drawer can firmly place 48 specimens, and the drawer can stand steadily on the lab work bench	89.4	
Operational friendliness	3	Sample drawer can be easily placed into the instrument	88.2	
	All barcodes on the primary sample tube can be scanned successfully	82.4
	The extraction kits can be correctly placed into three decks in the instrument	82.9
	The PCR reagent (premix) can be correctly placed into the instrument	90.3
	The consumable (spin tip) can be correctly placed into the instrument	88.9
	The consumable (pipette tip) can be correctly placed into the instrument	91.3
	The consumable (storage tube) can be correctly placed into the instrument	87.8
	The consumable (qPCR cap) can be correctly placed into the instrument	81.9
	The consumable (qPCR vial) can be correctly placed into the instrument	91.1
	The alarm for work completion can be obviously noticed	88.8
	Waste channel and trash bag can be easily installed to instrument	89.3
	Ease of GUI setup	14	The independent function can be clearly chosen and setup in the GUI	87.4
		The decontamination (UV light/HEPA) function can be clearly chosen and turned on in the GUI	87.3
The cooling function can be clearly chosen and setup in the GUI	82.8
The experimental result can be output to user's laptop	86.5
The new protocol can be imported to the instrument	86.5
Overall AVG			87.6	

**Supplemental Figure S1. Inter-instrument reproducibility of Ct values for pathogen**
**targets.**

Reproducibility testing was performed using inactivated pathogen controls across three
instruments (serial numbers 69–71), with five replicates per day for five days. Scatter
plots display Ct values for: S1a. SARS-CoV-2 (FAM), S1b. Influenza A (HEX), S1c.
RSV (Texas Red), S1d. Influenza B (Cy5), S1e. *Chlamydia trachomatis* (FAM), S1f.
*Neisseria gonorrhoeae* (HEX). Each color represents one instrument; error bars indicate
± 1 SD. Results confirm low variability and strong inter-instrument consistency.

(S1a)

(S1b)

(S1c)

(S1d)

(S1e)

(S1f)

521

[revised manuscript text omitted]

comprehensively evaluated the system's analytical and clinical performance, as well as
its usability in real-world laboratory settings.

Our findings demonstrate that the GenXplore[®] q4800 provides high precision and
stability across all optical detection channels, with coefficients of variation (CVs) for
both repeatability and reproducibility consistently below 5%. The system also showed
excellent analytical sensitivity, achieving lower limits of detection (LoDs) for SARS-
CoV-2, Chlamydia trachomatis (CT), and Neisseria gonorrhoeae (NG) compared to those
reported in the manufacturers' instructions for use (IFUs). Although the analytical
sensitivity evaluation was based on three replicates—fewer than the 20 replicates
recommended by CLSI EP17-A2—the qPCR kits used were CE-IVD certified, indicating
that their LoDs had been previously validated while reagent product registration. Given
the high reproducibility demonstrated by the GenXplore[®] q4800, we believe the observed
sensitivity reflects reliable system performance when integrated with certified reagents.

No cross-contamination was observed, even under high-throughput conditions
specifically designed to challenge the system's workflow integrity. In the inter-
instrument reproducibility study, three pathogen controls tested on instrument serial no.
69 showed slightly lower Ct values (Figure S1d–S1f). However, the internal control
results exhibited coefficients of variation (CVs) below 2%, indicating high system
stability. The slight Ct shifts are therefore likely attributable to control-specific batch
effects rather than instrument-related variability. Importantly, the observed differences—
within an inter-instrument CV of less than 5%—are not expected to impact diagnostic
interpretation.

[revised manuscript text omitted]

1c. Liquid handling module with high dispensing accuracy (Pipette tip 50 μ L: CV \leq 10%
at 5 μ L; CV \leq 2% at 50 μ L. Pipette tip 1000 μ L: CV \leq 10% at 10 μ L; CV \leq 2% at 100 μ L;
CV \leq 1% at 1000 μ L) and high dispensing precision (Pipette tip 50 μ L: CV \leq 1.5% at 5
μ L; CV \leq 0.75% at 50 μ L. Pipette tip 1000 μ L: CV \leq 3.5% at 10 μ L; CV \leq 0.75% at 100
μ L; CV \leq 0.75% at 1000 μ L)

1d. Graphical user interface (GUI), designed for flexible protocol setup and intuitive
operation.

1e. Optical detection module supporting fluorescence detection across multiple channels:
FAM, HEX/VIC/JOE, ROX/Texas Red, Cy5 and Cy5.5/Quasar 705.

1f. Thermal cycler module, with temperature accuracy $\leq \pm 0.4$ $^{\circ}$ C, uniformity $\leq \pm 0.4$ $^{\circ}$ C,
heating ramp rate ≥ 4 $^{\circ}$ C/s, and cooling ramp rate ≥ 3 $^{\circ}$ C/s.

A built-in barcode scanner enables automatic sample ID recognition. The system
accommodates 48 primary samples per run and 64 downstream qPCR reactions, with 16
additional wells available for controls or standards.

**Figure 2. Repeatability and reproducibility performance of GenXplore® q4800.**

2a. Repeatability performance across five optical channels. The SARS-CoV-
2/FluA/FluB/RSV Control (Vircell S.L.), containing four inactivated respiratory viruses
and one internal control, was tested with the corresponding qPCR kit in the GenXplore®
q4800 over 20 days (40 runs, 2 runs/day). The fluorescence signals were detected across
five channels: FAM (SARS-CoV-2, green box), HEX (Influenza A, yellow box), Texas
Red (RSV, orange box), Cy5 (Influenza B, red box), and Quasar 705 (internal control,
blue box). Box plots show Ct value distributions for each optical channel, demonstrating
low variability across 20 days.

2b & 2c. Inter-instrument reproducibility of internal control Ct values. Reproducibility
was assessed using three instruments (serial numbers 69–71) over five days, with five
replicates per day. Internal control Ct values from two qPCR kits were plotted for each
instrument: serial no. 69 (blue), 70 (orange), and 71 (gray). Error bars represent ± 1 SD.
The data confirm excellent consistency among instruments.

(2a)

(2b)

(2c)

**Figure 3. Analytical sensitivity of GenXplore® q4800 for detecting SARS-CoV-2, CT,**
 **and NG.**

Three inactivated pathogen controls (SARS-CoV-2, *Chlamydia trachomatis*, and
 *Neisseria gonorrhoeae*) were tested across four concentrations in triplicate over three
 399 days. 3a. LoD of SARS-CoV-2 met IFU claims (4 copies/reaction). 3b. LoD of CT met
 IFU claims (13 copies/reaction). 3c. LoD of NG met IFU claims (1.8 copies/reaction).
 The data demonstrate high sensitivity for low-copy pathogen detection.

(3a)

(3b)

(3c)

**Figure 4. Workflow layout in GenXplore® q4800 for contamination assessment and**
**final qPCR results.**

4a. Layout of the system's workstation. The possible area causing contamination consists
of extraction deck, sample drawer, reagent deck, storage deck and PCR setup deck.

4b. Movement path of the automated pipettor from the sample drawer to the extraction
deck (column 1) and the movement path of the nucleic acid extraction module, including
lysis, wash, and elution steps. Color grid: Light blue = 1 ng/ μ L DNA; Dark blue = 10
427 ng/ μ L DNA; Gray = ddH₂O.

4c. Movement path of the automated pipettor from elution wells to storage tubes to
transfer eluate.

4d. Movement path of the automated pipettor from storage tubes to PCR setup deck. 15
μ L of premix and DNA template will be dispensed into qPCR vials. Storage sample 1 to
16 will be dispensed into qPCR vial 1 to 16 and qPCR vial 49 to 64. The red frame
indicates mapping between storage tubes and qPCR vials.

4e. Dark blue grids indicate qPCR vials containing high-concentration (50ng/vial) DNA;
white numbers represent Ct values. Gray grids indicate no-template control qPCR vials.
All 32 positive vials showed successful amplification (mean Ct: 19.7, CV: 1.27%), while
all 32 no-template vials were negative, confirming zero cross-contamination.

(4a)

(4b)

(4c)

(4d)

(4e)

1	9	17	25	33	41	49	57
19.4	-	19.9	-	19.5	-	19.7	-

2	10	18	26	34	42	50	58
-	19.7	-	19.7	-	20.4	-	19.9

3	11	19	27	35	43	51	59
19.4	-	19.5	-	19.4	-	19.6	-

4	12	20	28	36	44	52	60
-	19.7	-	19.5	-	19.5	-	20.0

5	13	21	29	37	45	53	61
19.5	-	19.5	-	20.0	-	20.1	-

6	14	22	30	38	46	54	62
-	19.7	-	19.6	-	20.0	-	19.8

7	15	23	31	39	47	55	63
19.5	-	19.4	-	20.0	-	19.8	-

8	16	24	32	40	48	56	64
-	19.7	-	19.7	-	19.4	-	20.0

**Table 1. Clinical performance comparison with reference method.**

Contingency tables for 120 urine specimens comparing GenXplore[®] q4800 results with a
 semi-automated reference system (MagXtract[®] 3200 combining with LightCycler[®] 96)
 for detecting: 1a. *Chlamydia trachomatis* (CT, FAM); 1b. *Neisseria gonorrhoeae* (NG,
 HEX); 1c. *Mycoplasma genitalium* (MG, ROX). In all three cases, GenXplore[®] q4800
 achieved 100% agreement with the reference method and a Cohen's kappa coefficient of
 1.0.

(1a)

CT (FAM)		MagXtract [®] 3200 with LightCycler [®] 96		
		Positive	Negative	Total
GenXplore [®] q4800	Positive	89	0	89
	Negative	0	31	31
	Total	89	31	120

(1b)

NG (HEX)		MagXtract [®] 3200 with LightCycler [®] 96		
		Positive	Negative	Total
GenXplore [®] q4800	Positive	21	0	21
	Negative	0	99	99
	Total	21	99	120

(1c)

MG (ROX)		MagXtract [®] 3200 with LightCycler [®] 96		
		Positive	Negative	Total
GenXplore [®] q4800	Positive	6	0	6
	Negative	0	114	114
	Total	6	114	120

**Table 2. Usability evaluation based on professional user feedback.**

Eighteen items covering access control, operational friendliness, and ease of GUI setup
 were scored by 16 users. Average satisfaction scores exceeded 80 across all categories,
 with an overall average of 87.6, indicating high system usability.

Group	Number	Items	Average Satisfied Scores	
Access control	1	The access interface only allows authorized personnel to login	93.5	
	2	Sample drawer can firmly place 48 specimens, and the drawer can stand steadily on the lab work bench	89.4	
Operational friendliness	3	Sample drawer can be easily placed into the instrument	88.2	
	All barcodes on the primary sample tube can be scanned successfully	82.4
	The extraction kits can be correctly placed into three decks in the instrument	82.9
	The PCR reagent (premix) can be correctly placed into the instrument	90.3
	The consumable (spin tip) can be correctly placed into the instrument	88.9
	The consumable (pipette tip) can be correctly placed into the instrument	91.3
	The consumable (storage tube) can be correctly placed into the instrument	87.8
	The consumable (qPCR cap) can be correctly placed into the instrument	81.9
	The consumable (qPCR vial) can be correctly placed into the instrument	91.1
	The alarm for work completion can be obviously noticed	88.8
	Waste channel and trash bag can be easily installed to instrument	89.3
	Ease of GUI setup	14	The independent function can be clearly chosen and setup in the GUI	87.4
		The decontamination (UV light/HEPA) function can be clearly chosen and turned on in the GUI	87.3
The cooling function can be clearly chosen and setup in the GUI	82.8
The experimental result can be output to user's laptop	86.5
The new protocol can be imported to the instrument	86.5
Overall AVG			87.6	

**Supplemental Figure S1. Inter-instrument reproducibility of Ct values for pathogen**
**targets.**

Reproducibility testing was performed using inactivated pathogen controls across three
instruments (serial numbers 69–71), with five replicates per day for five days. Scatter
plots display Ct values for: S1a. SARS-CoV-2 (FAM), S1b. Influenza A (HEX), S1c.
RSV (Texas Red), S1d. Influenza B (Cy5), S1e. *Chlamydia trachomatis* (FAM), S1f.
*Neisseria gonorrhoeae* (HEX). Each color represents one instrument; error bars indicate
± 1 SD. Results confirm low variability and strong inter-instrument consistency.

(S1a)

(S1b)

(S1c)

(S1d)

(S1e)

(S1f)

521

The authors evaluated, a newly developed open-platform, fully automated nucleic acid testing system, for analytical and clinical performance using third-party molecular diagnostic kits targeting sexually transmitted infections and respiratory viruses. The study is well structured, with methodology aligned to CLSI standards and appropriate statistical analysis, including PPA, NPA, and Cohen's kappa. The work demonstrates strong analytical reliability with excellent repeatability, reproducibility, sensitivity, and absence of cross-contamination. Clinical validation showed perfect agreement with a semi-automated reference method, confirming diagnostic accuracy. The inclusion of usability testing adds practical relevance, and overall presentation is clear and systematic.

However, the evaluation could have been strengthened with broader validation. The pathogen panel (CT, NG, MG) and specimen type (urine) are somewhat limited, and the small number of *Mycoplasma genitalium*-positive samples reduces statistical power. The contamination control assessment, although showing no observed contamination in a single checkerboard setup, could have been more robust with additional runs or inter-instrument studies. Also, analytical sensitivity (LoD) testing involved fewer replicates than recommended by CLSI EP17-A2, and a higher replicate number per dilution would have provided stronger performance validation. Inclusion of additional pathogens, specimen types, and standardized reagent timing would have further enhanced the generalizability and comparative value.

The following aspects should be addressed to strengthen the manuscript:

1. A clear conflict of interest and funding statement should be included, given all authors are affiliated with the instrument development center.
2. The usability assessment involved 16 professional users, but the manuscript does not specify whether these participants were independent, had prior experience with the system, or were affiliated with the manufacturer. Clarifying these aspects would strengthen the credibility of the user satisfaction findings.
3. The Discussion could be strengthened by briefly comparing the reported run time (3–4.5 hours) and overall cost considerations with established closed systems, to better contextualize the platform's practical advantages and limitations.
4. The manuscript specifies that the system accommodates 48 samples per run. However, it does not indicate whether overlapping runs, consumable tracking, or random-access loading are supported. A brief clarification of workflow flexibility would help better understand throughput capacity in practical settings

Editor		
No.	Comments	Responses
1	Discussion: Lines 225-226: "The closed, all-in-one, nucleic acid testing systems have been developed and commercialized by major diagnostic companies in recent years. ": There are no references to support this statement.	Thank you for pointing this out. We have added an appropriate supporting reference, now cited as Reference 20 in the revised manuscript.

Reviewer #1		
No.	Comments	Responses
1-1	Although this is a novel study, it requires more data regarding the other extraction methods	We appreciate the reviewer's suggestion. In this study, we focused on validating the GenXplore® q4800 using two commercially available magnetic bead-based extraction kits: one from TANBead and one from MACHEREY-NAGEL. The extraction module integrates a whirl-spin bearing, heating strip, and magnetic rod to enhance lysis, bead collection, washing, and elution efficiency. This description has been added in Paragraph 1 on Page 4.
1-2	References are missing in the discussion part. Kindly elaborate on it	Thank you for this important observation. We have reviewed the Discussion section and added relevant peer-reviewed references in Paragraphs 1 and 3 on Page 9.

Reviewer #3		
No.	Comments	Responses
3-1	However, the evaluation could have been strengthened with broader validation. The pathogen panel (CT, NG, MG) and specimen type (urine) are somewhat limited, and the small number of Mycoplasma genitalium-positive samples reduces statistical power.	We agree with the reviewer. The current study focuses on initial performance validation. Future studies will expand the GenXplore® q4800 application to include diverse specimen types and additional pathogens. This limitation has been noted in Paragraph 4 on Page 10. Due to the limited number of Mycoplasma genitalium–positive specimens, we have removed this pathogen from the Abstract.
3-2	The contamination control assessment,	We appreciate the suggestion. In addition to

	although showing no observed contamination in a single checkerboard setup, could have been more robust with additional runs or inter-instrument studies.	the checkerboard test, we included results from extraction and qPCR negative controls across multiple experiments (repeatability, reproducibility, and analytical sensitivity) to assess cross-contamination. The methods and results are provided in Paragraph 3 on Page 5, in Paragraph 2 on Page 8 and Paragraph 1 on Page 10. We also deployed the GenXplore® q4800 (Serial No. 74) to a potential customer in Europe for beta-site validation. They conducted multiple cross-contamination assessments (checkerboard tests) using their own high-concentration DNA materials and qPCR kits (Ct 15–16, ~10⁷ copies) and reported no contamination. However, due to a non-disclosure agreement, these data cannot be included in the manuscript.
3-3	Also, analytical sensitivity (LoD) testing involved fewer replicates than recommended by CLSI EP17-A2, and a higher replicate number per dilution would have provided stronger performance validation.	We appreciate this comment. The LoD experiments were conducted with three replicates using CE-IVD–certified qPCR kits. These kits have validated LoD under regulatory requirements. Additionally, we demonstrated high repeatability and reproducibility, including in high-Ct ranges (30–35). We recognize the benefit of larger replicate numbers and have added this limitation in Paragraphs 3 and 4 on Pages 9 and 10, respectively.
3-4	Inclusion of additional pathogens, specimen types, and standardized reagent timing would have further enhanced the generalizability and comparative value.	We agree and have included this in our future work plans. This limitation is now described in Paragraph 4 on Page 10.
3-5	A clear conflict of interest and funding statement should be included, given all authors are affiliated with the instrument development center.	Thank you. A conflict of interest and funding statement has been added on Page 11, in accordance with journal policy.
3-6	The usability assessment involved 16 professional users, but the manuscript does not specify whether these participants were independent, had prior experience	Thank you for this request. The 16 participants were independent professionals affiliated with healthcare institutions or potential distribution partners. Most had prior

	with the system, or were affiliated with the manufacturer. Clarifying these aspects would strengthen the credibility of the user satisfaction findings.	experience with similar automated platforms. Their decision to collaborate was based on a comprehensive evaluation of system performance and usability. This information has been added to Page 8.
3-7	The Discussion could be strengthened by briefly comparing the reported run time (3-4.5 hours) and overall cost considerations with established closed systems, to better contextualize the platform's practical advantages and limitations.	We appreciate this excellent suggestion. We have included a comparison of processing time between the GenXplore® q4800 and systems such as Alinity m, cobas® 6800, Panther, and BD MAX in Paragraph 2 on Page 9.
3-8	The manuscript specifies that the system accommodates 48 samples per run. However, it does not indicate whether overlapping runs, consumable tracking, or random-access loading are supported. A brief clarification of workflow flexibility would help better understand throughput capacity in practical settings	We thank the reviewer for this practical suggestion. The GenXplore® q4800 supports overlapping runs and includes consumable and reagent traceability via barcode scanning. It is also compatible with laboratory information systems (LIS). These capabilities have been described in Paragraph 1 on Page 7.

Reviewer #4		
No.	Comments	Responses
4-1	The authors have mentioned that the repeatability studies were carried out by the use of inactivated controls for various pathogens. The authors are requested to clarify whether these controls were sequentially processed through all the six functional modules of the system.	Yes, all inactivated controls in the repeatability study were processed through the full GenXplore® q4800 workflow, including all six functional modules. This has been clarified in Paragraph 1 on Page 4.
4-2	The authors have mentioned that one of the criteria to analyze the reproducibility was to calculate the percentage CV of the internal control between instruments, however, the details of the internal control has not been provided.	Thank you for pointing this out. Both controls included human cellular material, and the qPCR kits are designed to detect the human RNase P gene as an internal control. This has been clarified in paragraph 3 in Page 4.
4-3	However, the laboratory would need to perform validation studies for each molecular diagnostic kit, to establish its performance characteristics, prior to use on patient samples, which may be beyond the	We agree this is a valid concern. For IVD users, validated system workflows may be provided through local distributors who are licensed to supply reagent kits. These distributors can offer pre-validated solutions

	capacity of many laboratories. The authors are requested to comment on how this issue can be addressed.	for the GenXplore® q4800. For LDT users, who typically lack distributor support, implementation requires collaboration to optimize automation parameters with in-house reagents. This has been discussed in Paragraphs 5 and 6 on Page 10.
--	--	---

Re: Spectrum03092-25R1 (Performance Evaluation of GenXplore® q4800: A Novel Open-Platform Automated Nucleic Acid Testing System)

Dear Mr. Ruei-Jiun Chi:

Your manuscript has been accepted, and I am forwarding it to the ASM production staff for publication. Your paper will first be checked to make sure all elements meet the technical requirements. ASM staff will contact you if anything needs to be revised before copyediting and production can begin. Otherwise, you will be notified when your proofs are ready to be viewed.

Sincerely,
Benjamin Liu
Editor
Microbiology Spectrum